# COVID-19: The Disease, the Immunological Challenges, the Treatment with Pharmaceuticals and Low-Dose Ionizing Radiation

**DOI:** 10.3390/cells10092212

**Published:** 2021-08-27

**Authors:** Jihang Yu, Edouard I. Azzam, Ashok B. Jadhav, Yi Wang

**Affiliations:** 1Radiobiology and Health, Isotopes, Radiobiology & Environment Directorate (IRED), Canadian Nuclear Laboratories (CNL), Chalk River, ON K0J 1J0, Canada; jihang.yu@cnl.ca (J.Y.); edouard.azzam@cnl.ca (E.I.A.); ashok.jadhav@cnl.ca (A.B.J.); 2Department of Biochemistry Microbiology and Immunology, Faculty of Medicine, University of Ottawa, Ottawa, ON K1H 8M5, Canada

**Keywords:** COVID-19, SARS-CoV-2, low-dose radiotherapy, X-ray

## Abstract

The year 2020 will be carved in the history books—with the proliferation of COVID-19 over the globe and with frontline health workers and basic scientists worldwide diligently fighting to alleviate life-threatening symptoms and curb the spread of the disease. Behind the shocking prevalence of death are countless families who lost loved ones. To these families and to humanity as a whole, the tallies are not irrelevant digits, but a motivation to develop effective strategies to save lives. However, at the onset of the pandemic, not many therapeutic choices were available besides supportive oxygen, anti-inflammatory dexamethasone, and antiviral remdesivir. Low-dose radiation (LDR), at a much lower dosage than applied in cancer treatment, re-emerged after a 75-year silence in its use in unresolved pneumonia, as a scientific interest with surprising effects in soothing the cytokine storm and other symptoms in severe COVID-19 patients. Here, we review the epidemiology, symptoms, immunological alterations, mutations, pharmaceuticals, and vaccine development of COVID-19, summarizing the history of X-ray irradiation in non-COVID diseases (especially pneumonia) and the currently registered clinical trials that apply LDR in treating COVID-19 patients. We discuss concerns, advantages, and disadvantages of LDR treatment and potential avenues that may provide empirical evidence supporting its potential use in defending against the pandemic.

## 1. Introduction

Humans have been living in a world affected by the novel coronavirus disease (COVID-19) for more than a year. Starting from a small group of infections, the highly infectious severe acute respiratory syndrome coronavirus 2 (SARS-CoV-2) viral strain has caused over 213 million reported infections, leading to ~4.46 million deaths so far [1]. Among the closed cases that had an outcome of either recovery and discharge or death, the case fatality rate is ~2% [1]. Although it has been argued that social measures have been largely ineffective [2], without the assistance of non-pharmaceutical interventions such as stay-at-home, use of sanitizers (hand washing), early case detection, and face masking, it is unlikely that the size of the outbreak can be mitigated or reduced [3]. Whereas active cases temporally declined in February and June 2021, they have rapidly risen again reaching an even higher record set in late April, and reaching over 744,000 daily new cases as of 19 August 2021 (among infected patients, 0.6% being in serious conditions) [1].

As an infectious disease, COVID-19 can be transmitted from human to human through respiratory droplets and aerosols created within a short distance, as well as prolonged interaction [4]. The World Health Organization (WHO) presented the “three Cs” as places with higher COVID-19 spreading risk: Crowded places with many people nearby, close-contact settings, where close-range conversations take place, and confined/enclosed spaces with poor ventilation [5]. These places include restaurants, bars, fitness facilities, theaters, and museums where people tend to be crowded and less fresh air is available. Thus, social distancing (at least 2 m apart and less than 30-min of close contact) has been advocated as a measure to reduce cluster transmission [6].

Similar to the seasonal flu, COVID-19 is a viral, respiratory illness. Whereas the flu results from an influenza virus infection, COVID-19 is caused by a new type of coronavirus—SARS-CoV-2 [7,8]. SARS-CoV-2 is one of the seven pathogenic coronaviruses that cause human respiratory diseases, which also include SARS-CoV, MERS-CoV (Middle-East respiratory syndrome coronavirus), HCoV-OC43 (human coronavirus OC43), HCoV-HKU1 (human coronavirus HKU1), HCoV-NL63 (human coronavirus NL63), and HCoV-229E (human coronavirus 229E) [9].

## 2. Epidemiology of COVID-19

Although the mortality data of COVID-19 are always compared with those of the flu as both induce severe respiratory symptoms that can lead to death, caution must be exercised when interpreting incidence. The Center for Disease Control (CDC) in the United States of America (USA) estimates the seasonal influenza mortality to be at ~0.1% over the past years [10]. In contrast to indirect counting as in the case of the flu, the CDC collects each COVID-19-related death report directly. Hence, researchers suggested that a comparison between the weekly deaths due to COVID-19 or the flu could be a more valid method [11]. For example, in the USA, the weekly flu deaths were around 500 in early 2020 and decreased at a later time in the year, whereas the weekly COVID-19 deaths soared to over 16,000 and fluctuated with time (Figure 1) [12]. This fluctuation may result from a shortage of ventilators and limitations associated with the work of health providers. 

Several situations complicate the statistics and treatment of COVID-19, including co-infection with SARS-CoV-2 and the common flu virus simultaneously, or infection with SARS-CoV-2 itself rendering existing diseases to deteriorate further and causing death [13]. Although asthma was associated with severe diseases during past influenza epidemics, and respiratory viruses were recognized as cause of acute exacerbations of asthma, scientific analyses did not correlate asthma with a higher severity or mortality in cases of COVID-19 [14]. As of early 2021, scientists still could not draw definitive conclusions on the correlation between asthma and COVID-19 outcomes due to various potential bias factors [15,16]. However, patients with dermatologic conditions or with gastrointestinal or liver ailments receiving drugs that affect the immune system may have atypical presentations of COVID-19, such as no fever [17,18,19]. Given the devastating influence of COVID-19 on the lung, chronic obstructive pulmonary disease (COPD) patients may have a higher risk of contracting COVID-19 with severe pneumonia and poor outcomes [20]. Compared to non-COPD patients, hospitalized COPD patients infected with the SARS-CoV-2 virus are 2.6 times more likely to experience admission to the intensive care unit (ICU), invasive mechanical ventilation, or death [21,22]. Although COVID-19 is a growing concern for COPD patients, the latter might be “protected” from severe consequences with appropriate inhaled corticosteroids [21,22]. For COVID-19 patients receiving systemic anticancer therapy, extra caution needs to be taken due to possible treatment-induced immunosuppression [18]. It has been considered that patients are at elevated risk of COVID-19-associated complications in scenarios that include diabetes, COPD, cardiovascular diseases, hypertension, malignancies, and infection with the human immunodeficiency virus (HIV). Patients who have received chemotherapy in the last 3 months, extensive radiotherapy, bone marrow or stem cell transplants in the last 6 months, or disruption of their immune system are also at increased risk [23,24].

## 3. Symptoms and Transmission of COVID-19

COVID-19 patients develop symptoms similar to those caused by the common flu, such as fever, cough, fatigue, body aches/muscle pain, headache, sore throat, shortness of breath, nausea, vomiting, or diarrhea [25,26,27]. One peculiar symptom that distinguishes COVID-19 from the flu is the loss of taste (dysgeusia) or smell (anosmia). Interestingly, this was not recognized as one of the criteria for testing until a significant portion of patients reported anosmia as the primary presenting symptom [28,29]. Notably, when a medical researcher who experienced a rapid onset of anosmia did not qualify for COVID-19 assessment, scientists signaled an alarm and persuaded decision-makers to include this symptom as one of the criteria [30].

As the absence or presence of COVID-19 signs and symptoms are not accurate enough for a reliable diagnosis, the standard method of diagnosis is real-time reverse transcription-polymerase chain reaction (RT-PCR), one of the most sensitive and accurate laboratory methods to detect viral ribonucleic acid (RNA) [31,32,33,34]. Upper respiratory specimens, including nasopharyngeal swab, oropharyngeal swab, nasal mid-turbinate specimen, anterior nares specimen, and saliva specimen, are collected for initial diagnostic testing [35]. In addition, as a traditional fast diagnostic procedure, chest computed tomography (CT) scans have been applied as a complement to RT-PCR for diagnosing COVID-19 due to its high sensitivity [36]. Typical and common CT findings include pulmonary parenchymal ground-glass opacities and consolidation, affecting at least one lobe [37]. In contrast to their sensitivity, the specificity of chest CT scans is relatively low—for instance, the CT findings overlap with signs of other infections, such as influenza, SARS, MERS, and other viral pneumonia [38,39]. Therefore, the CDC and ACR (American College of Radiology) do not recommend chest imaging as a first-line test or routine screening for COVID-19 as it is insufficient for diagnosis [40]. CT scans were recommended only for hospitalized, symptomatic patients with clinical indications for imaging.

The vast majority of COVID-19 patients experience mild or moderate symptoms—ranging from no symptoms to cold-like symptoms and to mild pneumonia. Some patients may have long-term effects, including neurological [41], cardiac, and respiratory harm [42], as well as mental health problems, and other potential challenges, such as business failure, unemployment, health inequality, social isolation, and even domestic abuse [43,44]. Unfortunately, 15% of COVID-19 patients develop severe symptoms (requiring oxygen) and 5% present with critical infections (requiring ventilation), and present with breathing difficulties, bluish face or lips, sudden confusion, serious pneumonia, and even respiratory failure [45]. In critical cases, patients end up with complications including acute respiratory distress syndrome (ARDS), sepsis, septic shock, and multi-organ failure (heart, kidney), and succumb to death [46,47,48]. It was also found that the infection was more likely to affect elderly individuals with chronic comorbidities, including hypertension, diabetes, cardiovascular disease, and cerebrovascular disease [46,49].

### 3.1. Transmission and Basic Reproduction Number 

The COVID-19 infection transmits easily through coughing, sneezing, singing, talking, and breathing heavily, as long as enough virus particles get into the other person’s mouth, nose, or eyes [5]. The incubation period between exposure and symptom onset ranges from 1 to 14 days, with a median time of 5 to 6 days [50]. Significantly, patients can shed the virus even 2 days before showing any symptoms, thus greatly enhancing the infection rate. However, the virus is most transmissible (viral shedding peak) when the RNA level is highest, which coincides with the onset of the symptoms [50].

One of the most discussed terms regarding the transmission capacity of COVID-19 is R0, or basic reproductive number, a mathematical and epidemiological metric to estimate the transmission potential of an infectious disease [51]. The term R0, pronounced “R naught”, originated from demography and ecology studies for calculating the number of offsprings born to one female in her entire life [52]. Epidemiologists then adopted the concept of R0 and defined it as the expected average number or a range of secondary cases in which one case would infect a completely susceptible population [53,54,55,56]. In general, if R0 is greater than 1 (R0 > 1), then each infected individual causes more than one new infection, indicating an outbreak or epidemic. If R0 is less than 1 (R0 < 1), then each infected individual causes less than one new infection, indicating a reduction or dying out of the disease [51,56,57]. In the context of COVID-19, R0 values range from 1.3 to 7.7 (which is more variable than the WHO-reported R0 range of 2–4), according to different variables, different methods for modeling, and different estimation procedures [51,58,59,60,61,62,63]. Specifically, an estimate of the mean R0 for COVID-19 in China for the period of January to April 2020 was 1.58–2.24, whereas the preliminary estimation of the mean R0 in January 2020 was 2.24–3.58, reflecting the possible effectiveness of lockdown and control strategies [62,64,65]. 

### 3.2. Additional Transmission Routes

Two additional possibilities for COVID-19 transmission attracted attention: Pre-symptomatic (virus detected before symptom onset) and asymptomatic (virus detected without symptom having developed) transmission [66]. One report claimed that 12.6% of cases might be transmitted during the pre-symptomatic period [67], and other models suggested that up to half of infections were caused by pre-symptomatic transmission [66,68]. Although the feasibility of actual transmission is still unclear [69], pre-symptomatic and asymptomatic transmissions, along with the majority of cases with mild symptoms, might contribute not only to disease sustainability but also to tolerance across populations. 

In addition to the pre-symptomatic transmission and asymptomatic transmission, the super-spreading event is another significant and challenging situation. Though a recent study shows that super-spreading may be largely driven by heterogeneity in contact behavior [70], and maybe overestimated due to biases in data collection [70]. Furthermore, patients can get re-infected on separate occasions, which means prior exposure to COVID-19 might not guarantee a hundred percent immunity to the virus whose genome is constantly mutating [71]. This also indicates that when the immune system fails to mount a protective response, viral reinfection, memory antibody levels reduction, and viral nucleic acids persistence could contribute to the second wave of virus replication and another positive test result [72,73]. 

## 4. The Structure and Pathogenesis of SARS-CoV-2

Coronaviruses are not newly identified pathogens. In 1968, an informal group of virologists recognized a new group of rounded shape viruses under the electron microscope. They named them coronavirus, recalling the characteristic appearance of the solar corona [74].

The structure of the coronavirus consists of enveloped, spherical, crown-shaped proteins covering the positive-sense, single-stranded RNA (+ssRNA). Its genome size is ~ 30 kb, classifying it as the largest of all RNA viruses [75]. It is thought that bats were the source of the initial coronavirus infections in humans and animals by potential cross-species transmission and spread [76]. Similar to other coronaviruses, the structural proteins of SARS-CoV-2 consist of a membrane (M) glycoprotein, an envelop (E) protein, a nucleocapsid (N) protein, and a spike (S) protein (Figure 2) [77]. The coronavirus particles are pleomorphic, without a defined structure, and the RNA is contained in the N protein [78,79]. The M glycoprotein, known as matrix protein, forms the outside layer of the virus, forming a connection between the membrane and the N protein through the transmembrane domain [78,79]. Rather than producing the lipid envelop by itself, the virus uses the host’s lipids for replication and morphogenesis [78,79]. The S protein, prominent on the viral envelop similar to a crown, is critical for the viral entry process [78,79].

Although most human coronavirus infections are mild, three coronavirus-induced outbreaks—SARS, MERS, and COVID-19—burst out in the last 20 years. Notably, the two viruses causing SARS and COVID-19, namely SARS-CoV and SARS-CoV-2, share the same route in infecting humans by binding to human virus entry receptor angiotensin-converting enzyme 2 (ACE2) (Figure 3) [80,81,82,83]. Research showed that the binding affinity of SARS-CoV-2 to ACE2 was similar to or even higher than the binding affinity of SARS-CoV to ACE2, consistent with the high structural similarity between SARS-CoV-2 and SARS-CoV [83,84,85,86]. Additionally, neuropilin-1 (NRP-1) and the tyrosine-protein kinase receptor UFO (AXL) were identified as receptors to facilitate SARS-CoV-2 infection of the human respiratory system [87,88,89,90]. The overall structural homology and high affinity to the same receptor of these two viruses may explain the high transmission rates of SARS and COVID-19 from human to human [84,85]. 

Similar to other coronaviruses, the SARS-CoV-2 entrance process consists of three steps (Figure 3) [91]:

1. Host cell attachment and receptor binding;

2. Host cell membrane and viral membrane fusion;

3. Viral genomic material residing in the host cell.

Coronaviruses employ a class 1 viral fusion protein, the homotrimeric S glycoprotein, which protrude from the viral surface to attach and bind to cell receptors [86,92,93,94,95]. SARS-CoV-2 S protein, in which a receptor-binding domain (RBD) structure is similar to that of SARS-CoV as it shares around 75% overall amino acid sequence identity, attracts ACE2 as a receptor to infect target cells [83,96,97,98]. ACE2 is not only highly expressed in lower respiratory tract cells such as type II alveolar cells (AT2) of the lung, but is found in organs such as the heart, kidney, and gastrointestinal tract, as well. This presumably contributes to COVID-19-induced ARDS, heart attack, kidney failure, and digestive syndromes [91,99]. During the infusion step, the S glycoprotein-ACE2 complex is proteolytically cleaved at a polybasic cleavage motif (PPAR) at the junction of two subunits of the spike protein (S1 and S2) by type 2 transmembrane serine protease (TMPRSS2) expressed on the surface of the host cell, leading to S protein activation and conformational alterations [100,101,102,103]. Similar to other coronaviruses, the S protein in SARS-CoV-2 is composed of two functional subunits, designated S1 and S2, of which the S1 subunit comprises the RBDs and stabilizes the prefusion state of the viral membrane-anchored S2 subunit [83,95,104]. Upon the dissociation of S1 from ACE2 by TMPRSS2, S2 undergoes a transition from a metastable pre-fusion state to a more stable post-fusion state, promoting the virus to enter target cells and release its genomic material into their cytoplasm [83,86,91,92,93,94]. Consistent with the sequencing map, S2 fusion machinery is more conserved than the S1 subunit, in line with the discovery that S1 is more exposed at the viral surface and is subject to more stringent pressure from the host immune system compared with the S2 subunit [83,104]. The sequence overall identity and structure conservation of non-RBD regions in the S protein such as the S2 subunit suggest potential targets for neutralizing antibodies and vaccines [83,86].

## 5. Mutations of SARS-CoV-2

Similar to other RNA viruses, SARS-CoV-2 may develop mutations in a relatively easier manner compared with DNA viruses, especially in the spike region due to genetic drift [85]. Genomic sequencing allows the identification and monitoring of changes in SARS-CoV-2, which helps in understanding and predicting the associated characteristics and their influence on public health [105]. The surveillance of emerging variants plays an important role in detecting viruses that spread rapidly, cause more severe symptoms, evade detection by diagnostic tests, reduce susceptibility to therapeutics with monoclonal antibodies, and escape natural or vaccine-induced immunity [105]. 

As early as March 2020, a D614G mutation was identified in the S glycoprotein of SARS-CoV-2, namely a single amino acid replacement of aspartic acid (D) with glycine (G) [106,107]. By April 2020, G614 has become the global dominant variant in the pandemic compared to the D614 form originally identified in the first human cases in Wuhan, China [106,108]. This variant is caused by an A (adenine) -to- G (guanine) nucleotide mutation at position 23,403 relative to the Wuhan reference strain, and is accompanied by three C (cytosine) -to-T (thymine) mutations: One in the 5’UTR (untranslated region) at position 241, one silent mutation at position 3307, and one missense mutation in RNA-dependent RNA polymerase (RdRp P323L) at position 14,408 [106,109]. G614 is associated with potentially higher viral load in COVID-19 patients and patients of younger age, but not with disease severity or mortality, compared with its counterpart D614 form [106,107].

As reported in November 2020, a unique variant of SARS-CoV-2 associated with farmed minks was identified in humans in Denmark [110]. This “Cluster 5” variant contains three substitutions (I692V and M1229I, and Y453F in RBD) and one deletion (HV 69–70 deletion, namely histidine and valine at positions 69 and 70, respectively) in the S protein that had not been observed previously [111]. Concerning the multiple mutations in this variant, there was worry that viruses can potentially reduce antibody neutralization in humans, leading to attenuated immune protection [111]. With extensive investigation and surveillance, no more cases related to the Cluster 5 variant have been observed since September 2020, suggesting that it is no longer circulating [110,111]. However, variants with Y453F (tyrosine substituted by phenylalanine at position 453) mutation on the S protein are still common in the Netherlands and have been reported sporadically in several other countries including the Russian Federation, South Africa, Switzerland, and the USA [111].

More complicated than the above variants, lineage B.1.1.7 (also known as 20B/501Y.V1) is a phylogenetic cluster that is widespread and outcompeting an existing population of circulating variants with a more transmissibility rate observed in the United Kingdom [108]. B.1.1.7 contains an unusually large number of mutations, particularly in the S protein, in which N501Y (asparagine replaced by tyrosine at position 501, one of the key contact residues in RBD) was correlated with potentially higher binding affinity to human and mouse ACE2 [112,113,114]. Globally, two other variants with multiple S protein mutations including N501Y but without HV 69–70 deletions have been circulating rapidly in Australia and South Africa [115,116]. In addition to N501Y, lineage B.1.1.7 was identified with other mutations, including thirteen non-synonymous mutations, three deletions, and six synonymous mutations which may be induced by chronic infection [112]. Since December 2020, N501Y has become the top variant of SARS-CoV-2 worldwide, identified in more than 1.1 million individual samples sequenced for COVID-19 virus and clinically recorded since its first detection in February 2020 [117,118]. Since June 2021, the T478K and L452R variants are growing quickly, culminating to 1 million in a total of the sequenced samples. These variants are among the top ten harboring RBD region mutations (the variants also include T478K, K417T, S494P, F490S, S477N, N439K, L452R, K417N, and E484K) [117,118]. 

## 6. The immunological Changes Associated with COVID-19

When the viral infection starts (i.e., when SARS-CoV-2 reaches the cells that line the respiratory tract), the host responds by triggering the immune system to fight back. Although most patients recover successfully, a few but a non-negligible proportion of infected patients develop severe symptoms characterized by severe pneumonia, ARDS, lymphopenia, exhausted lymphocytes, cytokine storm, and possibly antibody-mediated enhancement of viral entry [47,119,120]. While the immune response attempts to defend against the infection, variable(s) yet to be clearly elucidated often counterbalance immune benefits leading patients to be hospitalized. Although the symptoms and outcomes are extraordinarily heterogeneous, understanding the relationship between disease severity and the various immunological changes that occur is under intense investigation [121].

In general, when humans are virally infected, specific cells produce interferons to block viruses from reproducing. Furthermore, the immune system marshals antibodies to target the invaders [122]. Specifically, IFN-γ (interferon-gamma)—a lymphokine secreted by activated T cells—directs and regulates the synthesis of IP-10 (IFN-γ-inducible protein-10), a protein involved in the inflammatory response [123]. IP-10 directs the recruitment of T-helper-1 (Th1) cells for both normal host defense against intracellular pathogens of the lung and acute and chronic inflammatory processes [124]. For example, studies in mice showed that a strong interferon response is crucial in the timely resolution of SARS-CoV and MERS-CoV infections. Alternatively, lethal inflammatory immune reactions are activated if the interferon response is delayed [125,126]. Significantly increased IP-10 levels have been shown in both SARS and MERS patients by independent groups over the past years [127,128].

In addition to the COVID-19 common symptoms such as fever and dry cough, bilateral patchy shadows, or ground-glass opacity on chest CT images, indicate lung damage. Together with CT scans that reveal damage to the lung, patients also present with high levels of IL-1B (interleukin-1B, an inflammatory cytokine), IFN-γ (T helper cytokine), IP-10, and MCP-1 (monocyte chemoattractant protein-1), which is similar to the association of these pro-inflammatory cytokines with pulmonary inflammation in SARS patients [47,129]. Depressed total lymphocytes counts (lymphopenia), prolonged prothrombin duration, and elevated lactate dehydrogenase were documented as the three most common abnormalities in hospitalized patients [46]. One of the early clinical reports in the city of Wuhan observed that compared to non-ICU patients, ICU patients had higher plasma levels of IL-2 (interleukin-2, a T helper cytokine), IL-7 (interleukin-7, a cytokine essential for lymphoid cell survival), IL-10 (interleukin-10, an anti-inflammatory cytokine), GCSF (granulocyte colony-stimulating factor), IP-10, MCP-1, MIP1α (macrophage inflammatory protein 1α, a small inducible cytokine involved in inflammatory responses), and TNF-α (tumor necrosis factor α, a pro-inflammatory cytokine) [47,130,131]. It was also found that white blood cell count, neutrophil count, the levels of D-dimer (indicator for blood vessel dysfunction and clot formation), blood urea, and creatinine levels continued to increase, while lymphocyte counts continued to decrease in non-surviving patients [46,132]. Despite COVID-19 being such a heterogeneous disease affecting patients with different ethnicities, ages, and underlying conditions, the triad of cytokines IP-10, IL-10, and IL-6 was identified to be severity-related, which might help in anticipating disease progression and the length of hospitalization [133]. 

## 7. Therapeutic Strategies

In general, three pathologic mechanisms might be associated with poor prognosis [46]:

Virus-induced neutrophilia related to the cytokine storm; 

The sustained inflammatory process related to activated coagulation;

Virus-caused hypoxia and shock related to multi-organ dysfunction. 

Hence, a cytokine storm, which is associated with high concentrations of pro-inflammatory cytokines in the plasma, is positively correlated with the severity of COVID-19 [47].

Several approaches to treat COVID-19 have been advocated. Symptomatic management and oxygen therapy are two mainstay clinical treatments [119]. Furthermore, in early studies, convalescent plasma transfusion showed an effective clinical outcome for patients with severe and critical symptoms. Whereas the preliminary data were promising, a retrospective study calculated that the effective rate was below 60% [134,135]. In a survey of 16,287 UK hospitalized patients, the convalescent plasma treatment made no significant improvement in the survival of patients who received the treatment compared to those who did not [136]. Numerous clinical trials have been undertaken to explore additional options to treat COVID-19, with ~7662 COVID-19 ongoing studies, among which 1382 studies focus on pharmaceuticals [137,138], as highlighted below. 

In a retrospective study of clinical treatment of hospitalized COVID-19 patients in early 2020, most patients received antiviral therapy (oseltamivir), many received antibacterial therapy (moxifloxacin, ceftriaxone, or azithromycin), and a few received glucocorticoid therapy when no specific treatment was recommended except supportive care in early 2020 [46]. Antibacterial drugs proved to be ineffective, and neither oseltamivir nor methylprednisolone (a strong corticosteroid) improved outcomes. Among those admitted into the ICU—patients who also often presented with hypertension, diabetes, cardiovascular disease, or cerebrovascular disease—high-flow oxygen therapy, non-invasive ventilation, and invasive ventilation (including extracorporeal membrane oxygenation as a rescue therapy) were applied according to specific individual situations [46]. 

During the early chaotic period of the COVID-19 pandemic, no antiviral drugs were efficacious until remdesivir showed limited improvement in shortening the recovery time in hospitalized adult patients [139]. Nonetheless, the cytokine storm-induced morbidity and mortality remained elevated [140]. Baricitinib, an inhibitor targeting the intracellular signaling pathway of cytokines, including IL-6, IL-10, IFN-γ, and GMCSF (granulocyte-macrophage colony-stimulating factor), improved patients’ quality of life. It also reduced conventional inflammatory parameters and IFN biomarkers (serum IP-10 level and 25-gene IFN score) in clinical trials treating monogenic IFN-mediated auto-inflammatory diseases [141]. Recently, the United States Food and Drug Administration (FDA), issued an emergency use of the anti-inflammatory drug baricitinib, in combination with the antiviral drug remdesivir, for the treatment of dysregulated inflammation for severe COVID-19 patients [142]. It was found that baricitinib plus remdesivir were superior to remdesivir alone in shortening the recovery time and improving clinical status with fewer serious adverse events in patients with COVID-19 [143]. 

Although the US FDA has authorized the emergency use of baricitinib in combination with remdesivir, additional investigation of treatment methods is needed due to the many confounding factors. The compensating effect of other key events in the involved molecular pathways is always a complicated issue for molecular targeting drugs. This is also the reason for clinical trials with various drug combinations, conducted in the hope of obtaining co-effective outcomes such as shortened recovery and reduced mortality. Considering the unbalanced immune conditions in severely ill patients, the mortality rate was not reduced significantly, although there was some improvement (Figure 4). 

## 8. Vaccine Development

As an important pandemic control measure, various groups began seeking a safe and effective vaccine for this highly transmissible disease once the SARS-CoV-2 genome sequence was released [144,145]. 

The classical vaccine platforms commonly used for humans include a whole-inactivated virus, live-attenuated virus, protein subunit, and virus-like particle [146]. These platforms successfully prevented polio, MMR (measles, mumps, and rubella), seasonal influenza, and human papillomavirus, respectively [146]. As of August 2021, four out of 20 vaccines under the WHO evaluation process have been based on classical platforms, such as inactivated antigens [147]. Traditional virus-based vaccine development methods, such as attenuated or inactivated pathogens, although effective, require a large number of active viruses that only biosafety level 3 laboratories can safely produce in the case of COVID-19 [148]. Rather than dependence on the ability to culture live viruses, sequence information of the viruses alone can initiate the development of next-generation vaccines, leading to highly adaptable platforms in facing pandemics such as COVID-19 [146]. The next-generation platforms include viral vector vaccines, nucleic acid-based vaccines (DNA or mRNA), and antigen-presenting cells. For these platforms, the vesicular stomatitis virus-based Ebola virus vaccine (rVSV-ZEBOV) is a representative example—a recombinant, replication-competent vaccine consisting of a VSV vector engineered to express the main glycoprotein of the Zaire Ebola virus [146,149,150]. According to the WHO, as of 13 August 2021, 110 COVID-19 vaccine candidates are in clinical development, among which the protein subunit platforms account for 34%, followed by non-replicating viral vector, DNA, inactivated virus, and RNA platforms (15%, 9%, 15%, and 16%, respectively) [151]. Interestingly, intramuscularly injectable candidate vaccines with two doses were found to be the preferred choice among the candidates in the pre-clinical phase, compared with vaccines with other dosages (one dose or three doses) and routes of administration (oral, subcutaneous, intradermal, or intranasal) [151]. 

Having been studied for more than three decades, the use of mRNA is not a novel approach for vaccine development [152,153]. The search for effective vaccine tools that can be quickly designed and applied with flexible adaptiveness to antigenic changes in circulating strains is constant, together with updated strategies to combat newly emerging viruses and potential pandemics [146]. Previous efforts to combat SARS and MERS outbreaks accelerated the understanding of the epidemiology, pathogenesis, and diagnosis of SARS-CoV-2, contributing to the accelerated development of therapies as well as vaccines [148,154]. Although the pandemic worsened for almost 1 year until the first COVID-19 vaccine was developed, the turnaround cycle of a mature product for such broad recipients has been lightning-fast compared with vaccines for measles (~10 years), hepatitis B (~15 years), and especially meningitis (~90 years) [145].

Late in 2020, two mRNA-based vaccines were sequentially granted emergency use authorization by the FDA: The Pfizer-BioNTech (on 11 December 2020) and Moderna (on 18 December 2020) COVID-19 vaccines [155,156]. Pfizer-BioNTech (a more than 170-year-old pharmaceutical conglomerate collaborating with a previously little-known German biotechnology company) developed a lipid nanoparticle-formulated, nucleoside-modified mRNA (modRNA) vaccine called BNT162b2, which encodes the membrane-anchored pre-fusion full length of SARS-CoV-2 S protein whose conformation is locked by two proline mutations [157]. Most vaccinated healthy adults in phase 1 and 2 clinical trials produced S-specific CD8+ and type 1 helper T-cells (Th1 CD4+) with a high serum level of IFN-γ [157]. Although another candidate called BNT162b1, which encodes a secreted trimerized SARS-CoV-2 receptor-binding domain, showed a similar immunogenicity profile (antibody responses), BNT162b2 elicited a milder systemic reactogenicity profile (such as fatigue, headache, and fever) than what BNT162b1 caused in recipients, particularly in the elderly [158]. This stimulated further phase 2 and 3 clinical trials with the BNT162b2 vaccine, with the two-dose regimen showing high efficacy irrespective of age, sex, ethnicity, body-mass index, and pre-existing comorbidities [159,160].

Similar to Pfizer, Moderna (a pioneer mRNA biotech company founded in 2010), in collaboration with the Vaccine Research Center of the US National Institute of Allergy and Infectious Diseases (NIAID), developed a lipid nanoparticle-encapsulated mRNA vaccine called mRNA-1273, which encodes a pre-fusion stabilized full-length SARS-CoV-2 S protein [161,162]. In preclinical studies in nonhuman primate models, mRNA-1273 induced Th1 CD4+ cells and interleukin-21-producing CD4 T follicular helper (Tfh) cell responses, with low or undetectable Th2 or CD8 T-cell responses [163]. In addition to detecting robust SARS-CoV-2 neutralizing activity, rapid protection of airways, and absence of pathologic alterations in the lung of nonhuman primates, a phase 3 clinical trial involving more than 30,000 volunteers proved the mRNA-1273 vaccine to be effective in preventing COVID-19 without safety concerns [163,164].

Almost simultaneously to Pfizer and Moderna, Oxford University and AstraZeneca developed a recombinant adenovirus vaccine for COVID-19 consisting of a replication-deficient chimpanzee adenoviral vector (ChAdOx1), which contains the gene that encodes the codon-optimized full length of SARS-CoV-2 S protein [165,166]. This vaccine was redesigned from ChAdOx1 MERS, and a single dose of the vaccine proved to be protective for nonhuman primates (rhesus macaques) against MERS-CoV-induced disease [166,167]. In preclinical studies, ChAdOx1 nCoV-19 vaccine elicited robust humoral and cellular immune responses in both mice (strong Th1 response with high levels of IFN-γ and TNF-α and low levels of IL-4 and IL-10) and rhesus macaques (significantly increased spike-specific antibodies after the second immunization and detectable T cell responses when challenged by S protein) [166]. Follow-up clinical trials showed that, compared to the ~95% efficacy of Pfizer-BioNTech (BNT162b2) and Moderna (mRNA-1273) vaccines, the overall efficacy of the Oxford-AstraZeneca COVID-19 vaccine (ChAdOx1 nCoV-19) was above 70%: 60% in the group receiving two standard doses (SD/SD cohort, 5 × 10^10^ viral particles per dose), and 90% in the group receiving a half-dose followed by a standard dose (LD/SD cohort), respectively [165]. Unlike the two previously mentioned vaccines, ChAdOx1 nCoV-19 is the first reported non-profit vaccine, committed to low-income and middle-income countries with a promised price of USD 2–3 per dose [168,169,170]. In particular, concerning transportation, the ChAdOx1 nCoV-19 vaccine can use routine refrigerated cold-chain delivery (2–8 °C), which is more affordable and practical compared with the ultra-low temperature freezers (−80 °C) required for mRNA-based vaccines [165,168]. On 15 February 2021, the WHO listed two additional versions of ChAdOx1 nCoV-19—produced in the Republic of Korea and in India, rolling them out for emergency use in more than 70 countries across six continents and heading to cover 142 countries [171,172]. Several European Union (EU) countries temporarily suspended the use of this vaccine due to reports of 37 thrombotic events, including 15 deep vein thrombosis cases and 22 pulmonary embolism cases, among over 17 million vaccinated individuals in the EU and UK [172,173]. The decision was taken even though the number of thrombotic events in vaccinated individuals is significantly lower than what naturally occurs in a similar size cohort. Considering that the vaccine was not designed to address medical issues other than preventing COVID-19, the WHO, along with other EU countries, have advised continuation of the use of ChAdOx1 nCoV-19-based vaccine [172]. 

Although viruses may sustain genetic drift, there is the confidence that the homogeneous protein produced by the spike gene construct would maintain most neutralization-sensitive epitopes [85]. Undoubtedly, functional studies on viral proteins can play an important role in antiviral drug screening and vaccine development with the assistance of next-generation platforms, which may permanently change our ability to respond to newly emerging viruses [96,146]. As for how long antibody protection induced by natural infection lasts, this is still unknown. Vaccination against SARS-CoV-2 regardless of antibody status is currently recommended. Follow-up time will inform how long the vaccine-induced antibody protection can last [174]. For detailed information on COVID-19 vaccines in clinical development, refer to Appendix A which is adjusted from the WHO document [151]. Although a few claim vaccination is ineffective for limiting contagion, especially considering the vaccine breakthrough by the Delta variant (B.1.617.2) [175,176,177,178,179], vaccination is still recommended for everyone (≥12 years old) to prevent COVID-19 and particularly to avoid hospitalization and death [180].

## 9. The History of Low-Dose Radiotherapy of Non-COVID-19 Pneumonia

Radiation therapy, which has been used to treat cancer for more than 100 years, is not new in disease management. After the discovery of X-rays in 1895 and the development of therapeutic methods in the early1900s, clinicians have used radiation to kill malignant cells [181]. Unlike the relatively high doses used to sterilize tumors, lower doses were also applied in treating non-cancerous diseases.

Low-dose radiotherapy, in which the total dose applied is significantly less than the dose used in cancer treatments, was historically a successful option for pneumonia before the discovery of penicillin [182,183]. Reports have suggested that an absorbed dose of 1 Gy, which is 50 to 100 times lower than the total doses used by oncologists, has anti-inflammatory effects [183,184]. As early as 1905, Musser and Edsall reported on their X-ray treatment of five patients with unresolved pneumonia, which shed the first light on the potential benefit of radiation in treating respiratory diseases [185]. Almost two decades later, Heidenhain and Fried observed a systematic inflammatory-blocking function of low-dose X-ray treatment, showing clinical utility in treating deeper penetrating infections such as pneumonia [186]. 

In most follow-up studies performed in the 1930s, the applied doses ranged from 0.3 to 0.5 Gy [187]. In 1936, Powell applied low-dose radiotherapy in a clinical study with 231 patients after his first success on a patient with a complete recovery from lobar pneumonia [188]. A concern is that Powell did not randomize his patients as his staff recognized near-immediate relief from respiratory distress and circulatory distress after treatment [183,187,188]. In a study that appeared soon after, Scott noted that the patients experienced initial relief of symptoms, followed by a reduction of fever [189]. Surprisingly, no record of any toxicity events in the treatment group was noted as the investigators monitored for adverse radiation effects [189]. However, it may be argued that longer follow-up periods may have revealed a different spectrum of health effects. In 1942, Rousseau reported a full clinical recovery in the majority of patients with atypical sulfanilamide non-responsive pneumonia treated with low-dose radiotherapy, noting that the patients were free from toxic side effects [190]. In the following year, Oppenheimer reported that presumed-viral pneumonia patients treated with X-rays also showed complete fever resolution and disappearance of pulmonary consolidations [191]. Details of the brief history of X-ray therapy on non-COVID-19 patients are listed in Table 1.

In short, the early reports suggest that low-dose radiotherapy can reduce mortality rate, alleviate symptoms rapidly, and substantially reduce the severity of pneumonia [183]. Although the reported toxic effects of low-dose radiotherapy were minimal, systematic, evidence-based reasoning for the mechanisms underlying the efficacy of low-dose radiotherapy is needed. The promising early studies did not last long with the discovery of penicillin shifting scientific focus and funding mechanisms on antibiotics. Furthermore, there were safety concerns about long-term adverse outcomes including cancer and other degenerative diseases. The silence on the efficacy of low-dose radiotherapy was broken after 80 years as the global pandemic brought scientists to consider again the anti-inflammatory effects of low-dose radiotherapy in attenuating the severity of life threatening symptoms of COVID-19 [192].

**Table 1 cells-10-02212-t001:** Brief History of X-ray Therapy on non-COVID-19 Pneumonia.

Time	Author	Location	Radiation Type	Dosage	Patients Number	Diseases/Conditions	Result	Adverse Radiation Effects	Reference
1905	Musser and Edsall	Pennsylvania	X-ray	N/A	5	Unresolved pneumonia	40% symptoms disappeared	N/A	[185]
1907	Edsall and Pemberton	Pennsylvania	X-ray	Various	2	Unresolved pneumonia	100% cured	N/A	[193]
1916	Quimby and Quimby	New York	X-ray	Not given	12	Chronic chest conditions	100% prompt benefit	N/A	[194]
1924	Heidenhain and Fried	Worms, Germany	X-ray	Low doses	243	Unresolved chronic bronchopneumonia	75% good and very good improvement	N/A	[186]
1925	Krost (cooperation with M.T. Blumenthal)	Chicago	X-ray	Small doses (*)	12	Unresolved pneumonia (children)	92% apparent benefit	None	[195]
1930	Merritt and McPeak	N/A	X-ray	N/A	7	Unresolved pneumonia	86% cured	N/A	[196]
1936	Powell	Houston, Texas	X-ray	N/A	47	Unresolved pneumonia (lobar?)	100% convalescence with 2.5% mortality	N/A	[188]
1938	Powell	Temple, Texas	X-ray	Small doses (!)	104	Lobar pneumonia	Mortality rate less than 5%	N/A	[197]
1938	Powell	Temple, Texas	X-ray	small doses (!)	30	Bronchopneumonia	Mortality reduced from 30% to 13%	N/A	[197]
1939	Powell	N/A	X-ray	N/A	N/A	Acute pneumonias	N/A	N/A	[198]
1939	Scott	Niagara Falls	X-ray	N/A	138	Acute Lobar pneumonia	80% cured	None	[189]
1942	Rousseau	North Carolina	X-ray	Small doses (@)	72	Atypical bronchopneumonia or lobular pneumonia	N/A	None	[190]
1942	Rousseau	North Carolina	X-ray	Small doses (@)	104	Acute pneumococcic lobar pneumonia	94% recovered	None	[190]
1942	Rousseau	North Carolina	X-ray	Small doses (@)	29	Sulfonamide non-responsible pneumococcic lobar pneumonia	76% recovered	None	[190]
1943	Oppenheimer	New Hampshire	X-ray	Low doses ($)	36	Interstital pneumonia (children)	92% rapid and consistent improvement	N/A	[199]
1943	Oppenheimer	N/A	X-ray	N/A	56	Presumed-viral pneumonia	80% cured	N/A	[191]

(*) The dose was 5 mA, spark gap 7.5 in, distance 8 in, filter 3 mm of aluminum (Al) and 4 mm of sole leather, and time 5 min [195]; (!) 250–350 roentgens of 0.3 Åunit of effective radiation (135 kVwith 3 mm Al filter) given anteriorly or posteriorly over an area a little larger than the involved portion of the lung [197]; (@) 120 kV, 3 mm Al filter at a single dose at intervals of 36 h, if necessary, for one, two, or three doses [190]; ($) 1/3 to 1/5 of recommended pneumonia treatment; 140 kV, 20 mA, 0.5 mm Cu plus 1 mm Al filtration, the dose varying from 20 to 90 roentgens measured on the skin, through portals 15 cm^2^ directed to the involved region with the inclusion of the hila; the anode to skin distance is 35 cm [199].

## 10. Low-Dose Radiation Treatment of COVID-19

Although COVID-19 pneumonia is a novel disease that is different from other types of ARDS, severe COVID-19-associated ARDS shares typical ARDS lung pathology such as diffuse alveolar damage and hyaline membrane formation [120,200]. As Prasanna et al. summarized, the general rational for low-dose radiation treatment of COVID-19 is its inhibition of the cytokine storm, which promotes pulmonary dysfunction and ultimately ARDS [201]. Inflammation is a dynamic and progressive process that is tightly associated with redox-modulated reactions [202]. When recruited to sites of inflammation, macrophages and neutrophils generate reactive species, including reactive oxygen and nitrogen species (ROS and RNS). With multiple pro-inflammatory cytokines and chemokines being secreted, the latter together with elevated levels of ROS and RNS deteriorate redox homeostasis, and further worsen the disease [203]. During the past two decades, research has revealed that low-dose radiation-mediated homeostasis is associated with enhanced cellular detoxification of ROS by a major antioxidant enzyme (manganese superoxide dismutase, MnSOD) within the mitochondria [204,205,206,207]. This adaptive protection of mitochondrial metabolic functions is thought to provide experimental and theoretical support for using low-dose radiation to limit virus replication [207]. Other antioxidants, including glutathione, were also shown to be increased following exposure to low doses of sparsely ionizing radiation such as X and γ rays [208,209]. Schaue et al. suggested that it might be difficult and challenging for patients with complicated conditions and advanced age to rebalance redox levels, and low-dose radiation treatment might be of clinical value with its broad suppression of various inflammatory, pro-oxidant pathways at multiple levels [203,210]. 

Among the over 6000 clinical trials on COVID-19, only 17 focus on the relative efficacy of low-dose radiation therapy [211]. In June 2020, clinicians at Emory University reported their phase I interim analysis of a small pilot trial to treat oxygen-dependent elderly patients with COVID-19 pneumonia. They found that four out of five patients rapidly improved their breathing within 24 h after a single fraction of 1.5 Gy of X-rays applied to both lungs [212]. The patients recovered to breathe room air (unmodified, ambient air with a typical oxygen concentration of 21%) after being on supplemental oxygen for an average of 1.5 days after whole-lung irradiation [212,213]. Thereafter, clinical trials conducted in Iran found improvements similar to those observed in the latter study. An 80% response rate was observed in severe COVID-19 patients after whole-lung irradiation with a single fraction of 0.5 Gy [214]. These preliminary observations support the hypothesis that radiation at doses of ~0.5 Gy modifies the immune reaction to COVID-19 pneumonia by activating anti-inflammatory M2 macrophages [215]. Following up on the safety outcomes of the phase I trial, the same clinicians at Emory University carried out a phase II trial exploring the efficacy of immunomodulatory, single fraction, whole-lung moderate-dose of 1.5 Gy for patients with SARS-CoV-2 hyper-inflammatory pneumonia [216]. They reported a significantly shorter time of 3 days to clinical recovery in the radiotherapy cohort compared to 12 days in the control cohort. They also noticed significantly reduced levels of the inflammatory biomarker CRP (C-reactive protein) and LDH (lactate dehydrogenase), which suggest modulatory effects on immune cells. In November 2020, another pilot study in India showed a 90% response rate for low-dose radiotherapy (0.7 Gy in a single fraction) in COVID-19 patients having moderate to severe disease, which further supports the feasibility and clinical effectiveness of low-dose radiotherapy [217]. Overall, all of these clinical trials reported no early toxic effect of low-dose radiation (LDR). However, in February 2021, a study from Switzerland reported an identical survival rate (63.6%) in both 1 Gy-whole-lung radiation treated and sham treated groups, although the standard treatment of dexamethasone to ICU patients may have masked the anti-inflammatory effects of low-dose radiation [218]. Details of current radiation treatment of COVID-19 patients are listed in Table 2 (completed trials) and Table 3 (ongoing trials).

Whereas reports show attenuation of COVID-19 symptoms with relatively low-dose radiotherapy, scientific and public concerns about long-term side effects are evident. There is worry that radiation exposure will induce long-term adverse outcomes, including bystander effects, oxidative stress, chronic inflammatory responses, cytotoxicity, genomic instability, and ultimately cancer and degenerative conditions [202,236]. In particular, current regulatory guidelines are based on a linear no-threshold (LNT) model that assumes exposure to any dose of ionizing radiation increases cancer risk. The LNT model is a practical operational framework used in *radiation protection* to ensure the safety of workers and members of the public, which includes those who are particularly vulnerable to the effects of ionizing radiation. However, from the point of view of *risk assessment*, several studies claim beneficial effects from low-dose radiation (reviewed in [237,238]) and show that low-dose radiation effects often differ from those induced by high-dose radiation, at least in the case of the low-linear-energy transfer types of radiation such as X and γ rays [239,240]. These studies reveal that in vitro exposure of cells maintained in culture and in vivo exposure of rodents to low doses of sparsely ionizing radiation (i.e., X and γ rays) modulate changes at the molecular level, and provide significant evidence supporting up-regulation of DNA repair, anti-oxidation, and expression of adaptive responses [202,241]. 

With respect to treatment for COVID-19, low-dose radiotherapy may have unique advantages emerging from its successful history in treating inflammatory diseases and unresolved pneumonia: 

1. Low-dose radiotherapy can target inflamed areas such as the lung specifically, whereas antiviral, anti-inflammatory drugs, and statins are systemic drugs that affect the whole body.

2. The benefits of low-dose radiotherapy appear to outweigh the risk of long-term effects in the case of seriously ill COVID-19 patients. 

3. Low-dose radiotherapy is less likely to induce drug-resistant mutation in the virus compared to anti-viral drugs [242]. 

It is generally accepted that radiation is a mild mutagen [243]. As an RNA virus, SARS-CoV-2 is characterized by a moderate to high mutation rate. Mutation hot spots have been identified during the current pandemic, indicating potential drug resistance [109]. Antiviral drugs, the research topic granted the 2018 Nobel Prize in Chemistry (for phage display) demonstrated that therapeutic protein engineering can be modified under selective pressure, to target viruses [244]. Conversely, based on the reaction of viruses to antiviral drugs, a more resistant group of viruses might be selected under target drug therapies.

Low-dose radiotherapy is a cost-effective treatment that is available in most hospitals. In contrast, the expense for molecular-targeting therapies is challenging for a large number of patients in both developed and developing countries. Considering economic reasons, it may be argued that low-dose radiotherapy would reach more patients than targeted therapies and with a reduced financial burden. In this context, portable low-dose radiation systems for bedside treatment, in ICU, of critically ill COVID-19 patients would be advantageous [245]. Such devices have the added benefit of keeping COVID-19 patients separate from other patients, especially vulnerable patients receiving radiation therapy in the same hospital, thereby minimizing the probability of infection [245].

## 11. Discussion

Although the clinical outcomes of low-dose radiotherapy for COVID-19 patients are promising, the specific mechanisms for its immunomodulatory effects remain unclear. Various mechanisms may be involved in reducing inflammation by ionizing radiation, including the induction of apoptosis in immune cells, secretion of anti-inflammatory factors, and the reduced function of macrophages [246]. To further elucidate the underlying mechanisms, in vitro studies may apply strategies to induce molecular modifications in cells interacting with the coronavirus. Furthermore, low-dose radiotherapy may induce alterations in the immune microenvironment. Using flow cytometry, identification of molecular markers on immune cells as well as other cells impacted by the virus upon irradiation might be informative. It will also be interesting to study whether apoptosis of cytokine-producing infiltrating cells is a substantial component, as a fast response in COVID-19 patients was noticed after low-dose radiotherapy [241]. It has been recognized that low-dose radiotherapy does not induce post-treatment pancytopenia or immunosuppression, which implies that low-dose radiotherapy might not worsen immune activation or slow viral clearance [216]. This proposed idea may be tested by applying low-dose radiotherapy on pseudovirus-infected cells that are co-cultured with immune cells, and by comparing the spectrum of changes in cell surface biomarkers by flow cytometry, as well as the speed of viral clearance by live-cell imaging using confocal microscopy. Specifically, qPCR or sequencing technology may be applied to analyze gene expression during the interaction between immune cells and viral particles. This may shed further light on the molecular pathways associated with healthy cells fighting a viral attack.

There is a concern that the anti-inflammatory effect of low-dose radiotherapy may not be strong enough to inhibit the cytokine storm [241]. To elucidate the mechanisms for further application, low-dose radiotherapy may be combined with FDA-approved anti-inflammatory drugs, molecular-targeting drugs, and anti-viral drugs, such as dexamethasone, baricitinib, and remdesivir [139,143,247]. If the combined therapy shows an additive effect at the cellular level, verifying the effect in animal studies is essential before obtaining approval for eventual administration to patients in serious conditions. It will be interesting to explore whether low-dose radiotherapy would improve the effectiveness of those top candidates for targeted treatment.

With respect to the concerns about adverse effects of radiation exposure, different dosages may be applied to determine the minimum therapeutic dose level that can be effective in relieving inflammation, and whether the radiation dose should be delivered as a single bolus or be fractionated. Many confounders may bias and limit the result regarding the long-term effect of low-dose radiotherapy on humans. Considering that not enough studies of low-dose radiotherapy are available on animal models, Prasanna et al. indicated that additional pre-clinical studies in non-human primates, hamsters, or mice are required before proceeding with low-dose radiation therapy in COVID-19 patients [201]. Hence, in vivo experiments using different animal models are necessary to understand the variables associated with radiation delivery (i.e., dose, dose-rate, fractionation) on attenuating inflammation [201]. The use of low-dose radiation as an adjuvant therapy was recommended by others, opening more avenue for multi-modal therapy for severe COVID patients [248]. With suitable experimental models, various parameters can be optimized for beneficial clinical effects [248].

In alignment with Schaue’s opinion, it is critical—although tricky—to identify the patients who are likely to benefit from radiation treatment and to set the time point early enough for them to receive treatment and recover [203]. Theoretically, the patients with a clinical and radiological diagnosis of severe bilateral interstitial pneumonia might benefit from low-dose radiation to the lungs. Dr. Jian-Jian Li, who advocated low-dose radiotherapy early in the COVID-19 pandemic, suggested the necessity of precise evaluation of the potential effects of low-dose radiation in control of the cytokine storm and ARDS, as well as the proper timing and dosage using animal models [207]. By identifying biomarkers associated with relief of the cytokine storm by low-dose radiation, and by comparing their correlation with clinical laboratory results, it becomes possible to provide a checklist of critical markers to triage patients when they first step into the hospital. Those who might develop severe disease and might benefit most with the least risk from low-dose radiotherapy can get immediate treatment and avoid deterioration of their condition. If pandemics are inevitable, by expanding research on therapeutic avenues for COVID-19 and understating the underlying mechanisms, we may become better prepared to defend against other novel viral infection.

## Figures and Tables

**Figure 1 cells-10-02212-f001:**
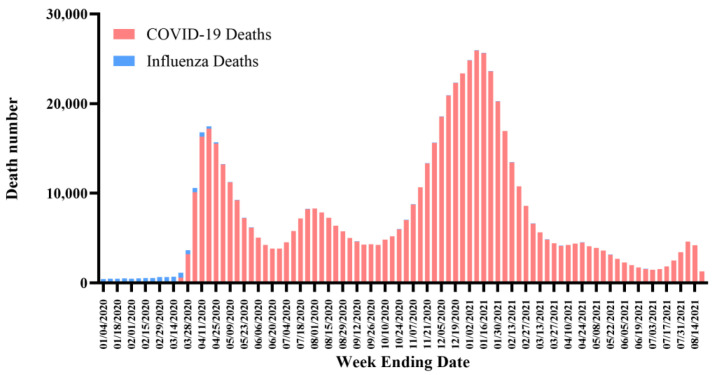
Weekly death analysis of COVID-19 and influenza in the USA. Data may lag by an average of 1–2 weeks, so the trend of decline in recent weeks may not accurately reflect the current situation. Available online: https://data.cdc.gov/NCHS/Provisional-COVID-19-Death-Counts-by-Week-Ending-D/r8kw-7aab (accessed on 24 August 2021).

**Figure 2 cells-10-02212-f002:**
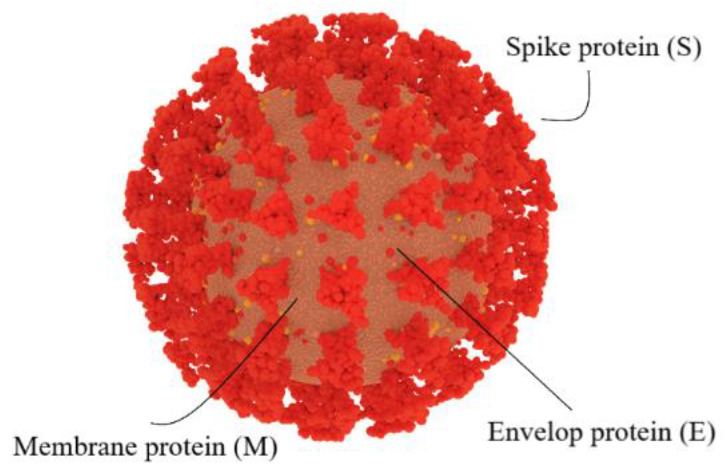
SARS-CoV-2 structure. Similar to other coronaviruses, SARS-CoV-2 contains a membrane (M) glycoprotein, an envelop (E) protein, a spike (S) protein, and a nucleocapsid (N) protein (not shown at the outer layer).

**Figure 3 cells-10-02212-f003:**
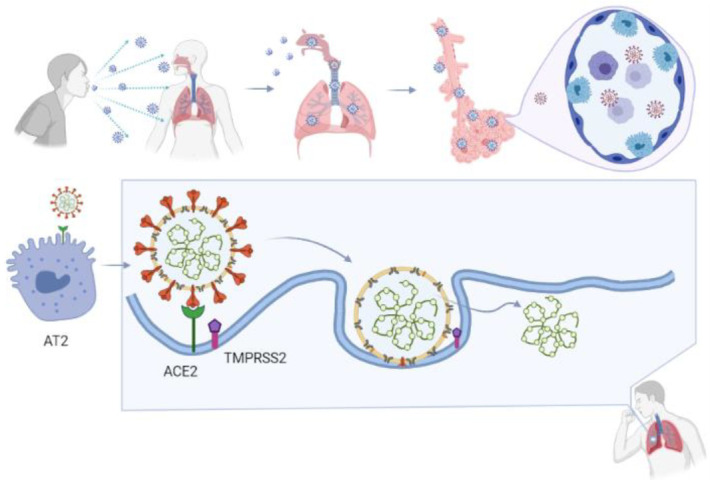
SARS-CoV-2 infection route. Infected COVID-19 patients can spread the virus by sneezing or coughing, or by any other route that drops the infectious particles onto another surface. When a person contracts the virus, the virus recognizes its binding target, ACE2, which is expressed on type II alveolar cells (AT2) in the lower respiratory tract. In the fusion step, the S protein on SARS-CoV-2 binds to ACE2, followed by cleavage of the S glycoprotein-ACE2 complex at polybasic motif (PPAR) at S1/S2 by type 2 transmembrane serine protease (TMPRSS2) and other proteases, leading to conformational alteration and activation of the S-glycoprotein. After fusion with the host cell membrane, the virus releases its genomic material into the cytoplasm.

**Figure 4 cells-10-02212-f004:**
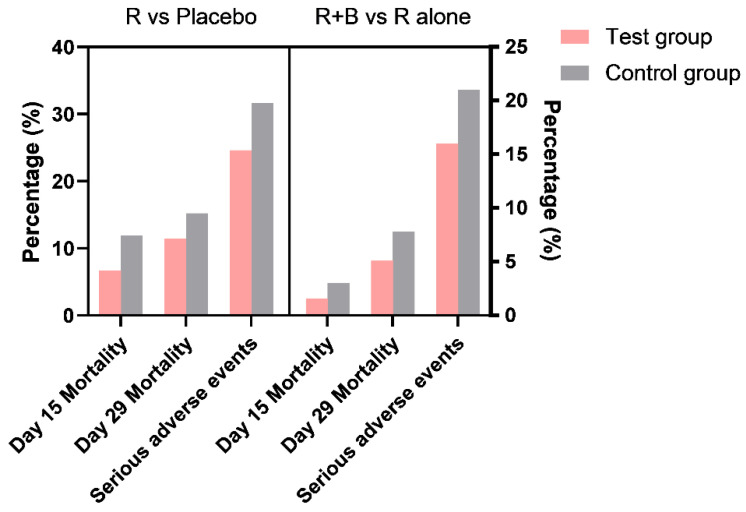
Remdesivir versus remdesivir plus baricitinib clinical trials. Comparison of mortality rate and serious adverse events of two clinical trials regarding remdesivir (R) compared with a placebo, and remdesivir in combination with baricitinib (R+B) compared with remdesivir alone (R alone). Data source: Remdesivir for the Treatment of COVID-19—Final Report [139], Baricitinib plus Remdesivir for Hospitalized Adults with COVID-19 [143].

**Table 2 cells-10-02212-t002:** Summary of Completed Clinical Trials Using Radiation in the Treatment of COVID-19.

ClincalTrial.gov ID	Acronym	Dosage	Fraction	Participants	Control	Median Age/Range	Female Percentage	Phase	Overall Recovery	Mean Time to Clinical Recovery	Mean Time to Discharge	No Acute Toxicities	Locations	Reference
NCT04366791	RESCUE 1–19	1.5 Gy	Single	10	No	90 (64–94)	80%	1	80%	1.5 days	12 days	No	USA	[212]
				20	10	76 (43–104)	55%	2	90%	3 days	12 days	No		[216]
NCT04394182	ULTRA-COVID	0.8 Gy	Single	15	No	80, 65	50%	1	100%	Various (*)	11 days	No	Spain	[219]
NCT04394793	N/A	70 cGy	Single	10	No	51 (38–63)	0%	N/A	90%	3–7 days	15 days	No	India	[217]
NCT04390412	N/A	0.5 Gy	Depends (!)	5	No	69 (60–84)	20%	1 and 2	75%	1 day	6 days	No	Iran	[214]
		0.5 Gy, 1 Gy	Depends (#)	10	No	75 (60–87)	20%		55.5%	N/A	8.7 days	No		[220]
NCT04598581	COVID-RT-01	1 Gy	Single	22	Sham irradiation	75 (54–84)	23%	2	73%($)	N/A	N/A	No	Switzerland	[218]
NCT04534790	N/A	1 Gy	Single	30	Not radiotherapy	N/A	N/A	N/A	N/A	N/A	N/A	N/A	Mexico	[221]
NCT04724538	N/A	99 mTc-pertechnetate aerosol	N/A	25	No intervention	N/A	N/A	1 and 2	N/A	N/A	N/A	N/A	Russian	[222]

(*) Patient 1: Day 5 after treatment; Patient 2: Day 2, 5, 7 less need, and dropping the intermittent oxygen support 2 months after treatment; improvement of symptoms such as asthenia and dyspnea after 48 h of treatment; (!) single fraction of 0.5 Gy, with another fraction of 0.5–1 Gy at least 72 h apart if necessary; (#) 5 received single-dose 0.5 Gy, one received two-dose 0.5 Gy, and four received single-dose 1 Gy; ($) 73% is the estimate of survival at 15 days in the low-dose radiation treatment group, whereas the 28-day survival estimates were the same at 63.6% in both low-dose radiation-treated group and sham-treated group.

**Table 3 cells-10-02212-t003:** Summary of Current Ongoing Clinical Trials Using Radiation Therapy for COVID-19.

ClincalTrial.gov ID.	Acronym	Dosage	Fraction	Participants	Control	Phase	Locations	Reference
NCT04433949	RESCUE1-19	≤ 1 Gy	Single	52	Not radiotherapy	3	US	[223]
NCT04414293	COVRTE-19	LDR	Single	41	No control group	2	Spain	[224]
NCT04427566	VENTED	80 cGy	Depends ($)	24	No control group	2	US	[225]
NCT04466683	PREVENT	35 cGy, 100 cGy	Single	100	Not radiotherapy	2	US	[226]
NCT04393948	N/A	100 cGy: single lung or whole lung	Single	48	Not radiotherapy	1 and 2	US	[227]
NCT04493294	N/A	LDR	Single	500	No control group	1 and 2	Switzerland	[228]
NCT04572412	N/A	50 cGy	Depends (^)	13	No control group	1	UK	[229,230]
NCT04377477	COLOR-19	70 cGy	Single	30	No control group	2	Italy	[231]
NCT04420390	LOWRAD-COV19	≤ 1 Gy	Single	41	No control group	N/A	Spain	[232]
NCT04380818	N/A	0.5 Gy	Depends (&)	106	Not radiotherapy	1 and 2	Spain	[233,234]
NCT04904783	LOCORAD	0.5 Gy	Single	20	Not radiotherapy	N/A	India	[235]

($) Eighty cGy single fraction, with a second option of 80 cGy if no improvement after 3–10 days; (^) single fraction of 50 cGy, with second option of 50 cGy, if necessary; (&) bilateral lung single dose LDR of 0.5 Gy, with option of additional 0.5 Gy after 48 h.

## Data Availability

Data source for Figure 1: https://data.cdc.gov/NCHS/Provisional-COVID-19-Death-Counts-by-Week-Ending-D/r8kw-7aab (accessed on 23 August 2021). Data source for Figure 4: Remdesivir for the Treatment of COVID-19—Final Report [139], Baricitinib plus Remdesivir for Hospitalized Adults with COVID-19 [143].

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
