# Peer review of "COVID-19: The Disease, the Immunological Challenges, the Treatment with Pharmaceuticals and Low-Dose Ionizing Radiation"

_cells, 2021, doi:10.3390/cells10092212_

Round 1

Reviewer 1 Report

Interesting paper covering the main steps of COVID 19.

Each part is well balanced and well written. The authors also focus on the use of low dose RT for treating pneumonia

I think that the author should mitigate their enthusiasm regarding RT, especially suggesting a potential instead of monoclonal antibodies because of RT cost-effectiveness and availability. However, I agree that monoclonal antibodies have a limit effect.

Minor comment:

figure 1: change the format, the week ending dates are hard to read.

Figure 2: typos, correct memberane to membrane

Pag 15 line 576 correct  In June of 2020 to In June 2020

Check reference 226: the paper on this study is recently published, I would suggest to change the table for this reference

Reviewer 2 Report

I think that the paper, despite obvious and evident limitations, can be accepted in the present form, given the great relevance being attributed to studies regarding COVID-19 by the Scientific Community.

- The Authors should refrain from using emphatic, moralistic or rhetoric tones through the paper.  

- In the Introduction section, the Authors state, “the mortality rate is ~ 2%”. This is not correct; with 4 million deaths, the mortality rate is 0.033%. If the Authors mean the COVID-19 lethality (i.e. case fatality rate or infection fatality rate), this is 0.15-0.20% in the global setting, and 0.1% in western countries (Ioannidis JPA. Global perspective of COVID-19 epidemiology for a full-cycle pandemic. Eur J Clin Invest. 2020 Dec;50(12):e13423. doi: 10.1111/eci.13423.) It is important to note that, in Italy, the median age of patients died after COVID-19 diagnosis was 82 years, which is close to average life expectancy in Italy (83 years).

- “With the assistance of public education, early detection, lock-32 down, travel ban, quarantine or isolation, and case monitoring, ~ 98% of patients diagnosed with COVID-19 successfully recover or are discharged from hospital”; this is non-sense, because the aforementioned measures should have the purpose of preventing infection, and not of impacting on the course of the disease. For instance, all these social measures have been proven largely ineffective (Bendavid E, Oh C, Bhattacharya J, Ioannidis JPA. Assessing mandatory stay-at-home and business closure effects on the spread of COVID-19. Eur J Clin Invest. 2021 Apr;51(4):e13484. doi: 10.1111/eci.13484.)

- When talking about vaccines, the Authors should clarify that their efficacy in terms of limiting contagion and reducing mortality has not been demonstrated:

Doshi P. Covid-19 vaccines: In the rush for regulatory approval, do we need more data? BMJ. 2021 May 18;373:n1244. doi: 10.1136/bmj.n1244.

Doshi P. Covid-19 vaccine trial protocols released. BMJ. 2020 Oct 21;371:m4058. doi: 10.1136/bmj.m4058.

Doshi P. Will covid-19 vaccines save lives? Current trials aren't designed to tell us. BMJ. 2020 Oct 21;371:m4037. doi: 10.1136/bmj.m4037.

Brown CM, Vostok J, Johnson H, Burns M, Gharpure R, Sami S, Sabo RT, Hall N, Foreman A, Schubert PL, Gallagher GR, Fink T, Madoff LC, Gabriel SB, MacInnis B, Park DJ, Siddle KJ, Harik V, Arvidson D, Brock-Fisher T, Dunn M, Kearns A, Laney AS. Outbreak of SARS-CoV-2 Infections, Including COVID-19 Vaccine Breakthrough Infections, Associated with Large Public Gatherings Barnstable County, Massachusetts, July 2021. MMWR Morb Mortal Wkly Rep. 2021 Aug 6;70(31):1059-1062. doi: 10.15585/mmwr.mm7031e2.

Predominance of Delta variant among the COVID-19 vaccinated and unvaccinated individuals, India, May 2021. J Infect. 2021 Aug 5:S0163-4453(21)00387-X. doi: 10.1016/j.jinf.2021.08.006.

- It is not clear how many “COVID-19 related deaths” are due to ARD or SARS, i.e. how many of these patients had a clinical and radiological diagnosis of severe bilateral interstitial pneumonia and could theoretically benefit from low dose radiation to the lungs. For example, regarding Italy, for 94% of patients died after COVID-19 diagnosis we do not have any clinical information.

- Low-dose radiotherapy for pneumonia was proposed when antibiotics were not available, now we have effective drugs for atypical-interstitial pneumonia (i.e. steroids and antibiotics), so exposing these patients to ionizing radiations without a clear advantage would be unethical. In addition, it is unclear which patients to select for X-ray treatment. The Authors should discuss these issues.
